# Cannabinoid Receptor Interacting Protein 1a (CRIP1a) in Health and Disease

**DOI:** 10.3390/biom10121609

**Published:** 2020-11-27

**Authors:** Emily E. Oliver, Erin K. Hughes, Meaghan K. Puckett, Rong Chen, W. Todd Lowther, Allyn C. Howlett

**Affiliations:** 1Department of Physiology and Pharmacology, Wake Forest School of Medicine, Medical Center Blvd, Winston-Salem, NC 20157, USA; eeoliver@wakehealth.edu (E.E.O.); ekhughes@wakehealth.edu (E.K.H.); mkpucket@wakehealth.edu (M.K.P.); rchen@wakehealth.edu (R.C.); 2Department of Biochemistry and Center for Structural Biology, Wake Forest School of Medicine, Winston-Salem, NC 20157, USA; tlowther@wakehealth.edu

**Keywords:** cancer, embryonic development, endocannabinoids, epilepsy, G protein-coupled receptors (GPCRs), hippocampus, retina, seizures, schizophrenia

## Abstract

Endocannabinoid signaling depends upon the CB_1_ and CB_2_ cannabinoid receptors, their endogenous ligands anandamide and 2-arachidonoylglycerol, and intracellular proteins that mediate responses via the C-terminal and other intracellular receptor domains. The CB_1_ receptor regulates and is regulated by associated G proteins predominantly of the Gi/o subtypes, β-arrestins 1 and 2, and the cannabinoid receptor-interacting protein 1a (CRIP1a). Evidence for a physiological role for CRIP1a is emerging as data regarding the cellular localization and function of CRIP1a are generated. Here we summarize the neuronal distribution and role of CRIP1a in endocannabinoid signaling, as well as discuss investigations linking CRIP1a to development, vision and hearing sensory systems, hippocampus and seizure regulation, and psychiatric disorders including schizophrenia. We also examine the genetic and epigenetic association of CRIP1a within a variety of cancer subtypes. This review provides evidence upon which to base future investigations on the function of CRIP1a in health and disease.

## 1. Introduction

The endocannabinoid system (ECS) includes CB_1_ and CB_2_ cannabinoid receptors and their endogenous agonist ligands. The CB_1_ cannabinoid receptor (CB_1_R) is expressed in great abundance in the central nervous system, where this G protein-coupled receptor (GPCR) serves a neuromodulatory function to reduce neurotransmitter release in both excitatory glutamatergic and inhibitory GABAergic neurons [1,2]. The CB_1_R is also expressed in many other cell types in the body, where it modulates energy storage, cardiovascular, intestinal, and endocrine/reproductive functions [3,4]. The endocannabinoid agonists anandamide (arachidonylethanolamide) and 2-arachidonoyl glycerol (2-AG) stimulate CB_1_Rs via orthosteric binding sites. CB_1_Rs also bind Δ^9^-tetrahydrocannabinol analogs, such as CP55940 or HU210, and non-cannabinoid ligands, such as WIN55212-2, which activate the receptor with greater potency and/or efficacy than do the endogenous ligands [5,6,7]. CB_1_Rs interact with several G proteins, predominantly the pertussis toxin-sensitive Gi/o family [8]. CB_1_-stimulated release of Gi/o α subunits leads to inhibition of adenylyl cyclase, which reduces cyclic AMP and protein kinase A (PKA)-mediated phosphorylation of its substrates [9] (Figure 1). Release of the βγ dimers leads to reduction in the currents across N, P and Q Ca^2+^ channels (Figure 1) and G protein-coupled inwardly-rectifying potassium channels (GIRKs) [10,11]. These βγ dimers also initiate src-mediated cascades whereby receptor tyrosine kinases activate mitogen-activated protein kinases (MAPKs) and focal adhesion kinases (FAKs) at sites of actin remodeling and integrin signals [12,13].

In addition to stimulation by small molecule agonists, the CB_1_R protein function is also regulated by protein-protein interactions [14,15,16]. The most obvious are the G proteins, which not only signal to effector molecules (adenylyl cyclase, ion channels), but also adjust the affinity of CB_1_R for agonists via a negative heterotropic interaction [17]. Cannabinoid receptor-interacting protein 1a (CRIP1a) alters the preference for Gi/o subtypes from signaling via Gi3 and Go to signaling via Gi1 and Gi2 [18] (Figure 2) and attenuates agonist-mediated internalization of CB_1_Rs [19,20,21] (Figure 3). Additional interactions involve the β-arrestins 1 and 2 which alter the cellular signaling patterns and desensitize the receptor-G protein interaction [22] (Figure 3). GPCR-associated sorting protein (GASP) directs internalized CB_1_R trafficking to the lysosome for degradation [23,24].

As reviewed herein, our understanding of how CRIP1a influences biochemical and cellular homeostasis has been advanced by experimentation using model cell systems, such as N18TG2 neuroblastoma cell clones expressing endogenous CRIP1a at wild-type, overexpression, or knockdown levels, and HEK293 fibroblast cells expressing exogenous CRIP1a at various levels. Studies are now beginning to report physiological and pathophysiological associations with CRIP1a. This review summarizes evidence that implicates CRIP1a in health and disease states.

## 2. CRIP1a Cellular Mechanisms of Action

CRIP1a was first identified by Niehaus and colleagues [20] using a yeast two-hybrid system to screen human brain cDNA using a bait specific to the C-terminal 55 amino acids of CB_1_R, excluding the fourth loop amino acids 400–417 encompassing helix 8. Further experiments determined that CRIP1a suppressed the CB_1_R-mediated inhibition of voltage-gated Ca^2+^ channels, as depicted in Figure 1. Exogenous expression of CB_1_R in superior cervical ganglion neurons suppressed the Ca^2+^ current, even in the absence of CB_1_R agonists (constitutive receptor activity). Addition of the inverse agonist rimonabant (also known as SR141716) reversed the CB_1_R-mediated suppression of Ca^2+^ currents, although to a lesser magnitude when CB_1_R was co-expressed with CRIP1a. These studies did not detect an effect of CRIP1a expression on agonist-stimulated CB_1_R suppression of the Ca^2+^ current [20]. One interpretation of these results is that CRIP1a can block the coupling of specific G proteins responsible for basal (constitutive) CB_1_R activity, but not block G proteins responsible for the agonist-stimulated response.

In order to investigate the mechanism by which CRIP1a suppressed CB_1_R constitutive activity, Blume and colleagues created stable CRIP1a knockdown and overexpression models using siRNA and CRIP1a cDNA transfection, respectively, in N18TG2 neuroblastoma cells [18]. Importantly, these cells natively express both CRIP1a and CB_1_R. Whole-cell lysate Western blotting and immunocytochemistry experiments revealed that CRIP1a expression was inversely related to CB_1_R surface levels without any change in overall CB_1_R concentration [18]. Because inhibition of protein synthesis by cycloheximide did not alter these results, the CRIP1a-mediated reduction in CB_1_R surface concentration is likely independent of de novo synthesis and instead utilizes an internalization or translocation mechanism. 

Using these knockdown and overexpression neuronal cell lines, Blume and colleagues discovered that CRIP1a reduced both constitutive and agonist-dependent activities of CB_1_R [18]. As depicted in Figure 1, basal phosphorylation of extracellular signal-regulated kinase (ERK)1/2 was dependent upon constitutive CB_1_R activity and was reduced in CRIP1a overexpression and increased in knockdown cells [18]. Thus, CRIP1a negatively modulated constitutive CB_1_R functions. CB_1_R agonists increased ERK1/2 phosphorylation above baseline, an effect that was blocked by a dynamin inhibitor dynasore [18], suggesting that this mechanism requires CB_1_R internalization. CP55940 (full agonist) evoked greater ERK1/2 phosphorylation in CRIP1a knockdown cells, although no change was observed in CRIP1a overexpressing cells. CRIP1a knockdown cells also exhibited enhanced CP55940-mediated attenuation of forskolin-stimulated cAMP accumulation, which was unaltered in overexpression cells [18]. These experiments identified a novel agonist-dependent role for CRIP1a in the suppression of CB_1_R activity that is agonist-selective. Endogenous levels of CRIP1a appear sufficient to achieve maximal effects in agonist-dependent signaling.

CRIP1a suppression of agonist-stimulated CB_1_R cellular signaling is unlikely to be a consequence of altered CB_1_R affinity for specific agonists, because CRIP1a expression failed to alter the ligand binding affinity for CB_1_R radioligand binding determinations using hCB_1_R-HEK cells [21]. Smith and colleagues found evidence for a mechanism by which CRIP1a influenced G protein activation by CB_1_R in an agonist-specific manner. CB_1_R-stimulated basal (constitutive) [^35^S]GTPγS binding to G proteins in hCB_1_R-HEK293 cell membranes was unaltered by CRIP1a overexpression. However, both the E_max_ of stimulation by high efficacy agonists (2-AG ether, HU210, WIN55212-2, CP55940) and maximal inhibition by an inverse agonist (rimonabant) were significantly attenuated by CRIP1a overexpression with no change in their apparent potency (EC_50_). Lower efficacy agonists (methanandamide, levonantradol, Δ^9^-tetrahydrocannabinol) were insensitive to CRIP1a overexpression. The specificity against low-efficacy agonists may be a consequence of cellular environment. The same experiment in N18TG2 neuroblastoma cell membranes showed that [^35^S]GTPγS binding stimulated by both WIN55212-2 (high-efficacy) and methanandamide (low-efficacy) was attenuated by CRIP1a overexpression [21]. Furthermore, CRIP1a was only capable of attenuating agonist activity in hCB_1_R-HEK293 cell membranes with high Na^+^ conditions (100 mM NaCl), and only capable of attenuating inverse agonist activity in the absence of Na^+^. Intriguingly, Na^+^-free conditions allowed CRIP1a to modestly attenuate otherwise unaffected basal [^35^S]GTPγS binding [21]. These experiments indicate that CRIP1a is capable of attenuating both constitutive and agonist-mediated CB_1_R signal transduction by reducing the maximal G protein activation, depicted in Figure 1, and furthermore that Na^+^ influences this mechanism and the sensitivity to agonists and inverse agonists.

Blume, Eldeeb and colleagues performed immune-selective scintillation proximity [^35^S]GTPγS-binding assays that implicate CRIP1a in G protein switching, which may underlie the modulation of CB_1_R activity [18]. These experiments revealed that CRIP1a overexpression eliminated both the CP55940-evoked increase and the rimonabant-evoked decrease in Gi3 and Go activation. Instead, CRIP1a overexpression elicited increased [^35^S]GTPγS binding to Gi1 and Gi2 in response to CP55940. CRIP1a knockdown potentiated the CP55940-evoked [^35^S]GTPγS binding and the rimonabant-evoked reduction in [^35^S]GTPγS binding to Gi3 and Go [18]. It is unclear how CRIP1a performs this subtype selection depicted in Figure 2, but three reasonable hypotheses include one model whereby CRIP1a binding to the C-terminal domain of the CB_1_R sterically hinders this domain from interaction with Gi3/o; a second model whereby CRIP1a binding sterically improves domain access by Gi1/2; a third model in which CRIP1a recruits Gi1/2 directly to CB_1_R for activation. Combined, these data suggest that CRIP1a can exert broad suppression of constitutive and agonist-dependent activities by selectively altering the G protein subtype accessible to CB_1_R and reducing total G protein activation in an agonist-selective manner.

Blume and colleagues [19] further investigated the influence of CRIP1a on CB_1_R plasma membrane trafficking in N18TG2 cells. CP55940 or WIN55212-2 (10 nM) reduced cell surface CB_1_R within minutes via a clathrin- and dynamin-dependent process. This agonist-stimulated loss of cell surface receptors was abrogated by CRIP1a overexpression [19], suggesting that CRIP1a attenuates CB_1_R internalization or sequestration. Prolonged CP55940 exposure culminated in a complete recovery of CB_1_R cell surface expression in parental cells within hours via a mechanism involving new protein synthesis. However, recovery for CRIP1a-overexpressing cells never returned to the same cell surface maximum as observed in parental N18TG2 cells. For prolonged WIN55212-2 exposure, recovery reached completion only in CRIP1a-knockdown cells. These studies demonstrate that CRIP1a inhibits agonist-stimulated CB_1_R cell surface internalization or sequestration, as depicted in Figure 3, but also suppresses re-establishment of steady state plasma membrane levels upon prolonged ligand occupancy [19].

Using a HEK293 transfection model and immunohistochemical tracking of cell surface receptors, Mascia and colleagues [25] confirmed that co-expression of CRIP1a with CB_1_R significantly attenuated internalization, but also inhibited internalization of metabotropic glutamate receptor 8a (mGlu8aR) from the surface. These researchers found that five out of nine amino acids in the distal C-terminus of CB_1_R (DTSxxAL) were required for the attenuation of CB_1_R internalization by CRIP1a, according to results of an alanine scanning procedure. A somewhat homologous sequence was found in the C-terminus of mGlu8aR (NTSxxKT), with some notable differences in amino acid properties [25]. Blume, Patten and colleagues [26] identified another site capable of binding CRIP1a located in the central CB_1_R C-terminal. Co-immunoprecipitation experiments using CRIP1a, CB_1_R, and either of two peptides designed to mimic the 14 distal residues (VTMSVSTDTSAEAL) or 19 central residues (TAQPLDNSMGDSDCLHKH) of the CB_1_R C-terminus revealed that both peptides significantly competed against the full-length CB_1_R for binding to CRIP1a [26]. These data describe the presence of two distinct binding sites within the CB_1_R CRIP1a-binding domain embedded in the C-terminal, containing several minimally required amino acids. This further reveals that CB_1_R is not the only receptor possessing CRIP1a binding potential.

Blume, Patten and colleagues [26] identified competitive interactions between CRIP1a versus β-arrestin for binding with CB_1_R at these sites of interest. Immunoprecipitations revealed CB_1_R could co-immunoprecipitate with either CRIP1a or β-arrestin1/2; CRIP1a and β-arrestin1/2 could not be co-immunoprecipitated with each other, however. Confocal microscope images identified the attenuation by CRIP1a of both CB_1_R internalization and β-arrestin recruitment in response to agonists [26]. Co-immunoprecipitation studies using the CB_1_R C-terminal-mimetic peptides (described above) elucidated the competition between CRIP1a and β-arrestin1/2. The distal C-terminal-mimetic peptide (VTMSVSTD**T**SAEAL) was predicted to bind CRIP1a from yeast 2-hybrid screens [20], whereas the central C-terminal-mimetic peptide (TAQPLDN**S**MGD**S**DCLHKH) was expected to bind β-arrestins [27,28,29]. However, both central and distal peptides served as common binding sites, competing for binding CRIP1a and β-arrestin1/2 in a phosphorylation-dependent manner. Phosphorylation of specific residues (bolded and underlined) attenuated the interaction with CRIP1a while promoting interaction with β-arrestin1 and/or 2 [26]. These findings imply that CRIP1a competes with β-arrestin for binding with the CB_1_R in a manner dependent upon the phosphorylation state of the CB_1_R, depicted in Figure 3. It is therefore reasonable to suggest that CRIP1a can modulate recruitment of β-arrestin to the CB_1_R.

## 3. CRIP1a Distribution in the Mammalian Brain

Defining the physiological and pathological significance of CRIP1a requires first identifying which regions and cell types of the central nervous system express the protein. Guggenhuber and colleagues [30] mapped CRIP1a and CB_1_R in the brains of adult male C57BL/6N mice using immunohistochemistry and in situ hybridization. The mRNA of both proteins was found in significant amounts in the forebrain and hippocampus with evidence of overlap. In the hippocampal formation, CRIP1a mRNA and protein were expressed at high levels in both glutamatergic and GABAergic neurons throughout the cornu ammonis (CA), hilum, and dentate gyrus (DG) [30] as depicted in Figure 4. Immunoreactive CB_1_R protein and mRNA have also been identified in glutamatergic pyramidal cells, glutamatergic hilar mossy cells, and GABAergic interneurons [30,31,32,33] (Figure 4). CB_1_R mRNA levels were abundant in GABAergic but modest in glutamatergic neurons, compared with the homogeneous distribution of CRIP1a mRNA [30,31]. The question of whether CB_1_Rs are expressed in granule cells, however, has not been fully resolved. Guggenhuber and colleagues argued that CRIP1a was notably expressed in CB_1_R-deficient granule cells [30] but defined granule cells by the soma-containing stratum granulosum layer which necessarily excludes granule cell axons and terminals (also called “mossy fibers” which are distinct from mossy cells) that project into distal layers (Figure 4). This is a common practice in studies mapping protein expression in regions such as the hippocampus, whereby whole cells are conflated with soma-containing tissue layers. While some studies support the assertion that granule cells are CB_1_R-negative [31,32], others provide evidence that granule cells can be CB_1_R-positive [34,35]. The granule cell problem is further complicated by the arguments that their mossy fibers are capable of both glutamate and GABA release (reviewed in [34]), making correct identification of synapses more difficult. Therefore, we simply do not have enough information to know the CB_1_R and CRIP1a co-localization in granule cells/mossy fibers, but there appear to be cellular regions in the DG that express CRIP1a without CB_1_R.

Immunoreactivity experiments performed by Guggenhuber and colleagues [30] revealed co-expression of CB_1_R and CRIP1a in CA pyramidal cells, CA interneurons, and DG hilar mossy cells, with varying levels of proximity (separated, close contact, and overlapping). Subcellularly, CRIP1a protein expression was predominantly restricted to presynaptic termini in both glutamatergic and GABAergic neurons [30], consistent with the established presynaptic membrane localization of CB_1_R (reviewed by [42]) (Figure 4). Unfortunately, no information was provided on dopaminergic or cholinergic innervation of the hippocampal formation, additional hippocampal cell types, nor distinguishing between GABAergic interneuron subtypes. These are excellent areas for further research as the hippocampal mapping of CRIP1a becomes more precise.

Immunohistochemical experiments from Smith and colleagues [21] mapped CB_1_R and CRIP1a within the cerebellum in adult male Sprague-Dawley rats and GAD67-GFP mice, demonstrating that CRIP1a and CB_1_R are co-expressed or independently expressed based upon the cell type. CRIP1a was expressed strongly in the CB_1_R-deficient granule cell layer and the CB_1_R-dense molecular layer near, but not within, the CB_1_R-positive perisomatic region of purkinje axon terminals. Co-staining with synaptic vesicle glycoprotein 2 (SV2), a presynaptic marker [43,44], supports that CRIP1a is expressed at presynaptic termini. Co-staining with glutamate decarboxylase (GAD67) reveals that CRIP1a is present in GABAergic inhibitory cells in the molecular layer, but not granule cell layer [21]. Combined findings from multiple brain regions support a model whereby CRIP1a is localized both at presynaptic regions with CB_1_R or at regions devoid of CB_1_R, suggest the intriguing hypothesis that CRIP1a can function via CB_1_R-independent mechanisms. The distribution of CRIP1a in other areas of the brain, such as the forebrain, have yet to be examined in detail.

## 4. CRIP1a in Development

Zheng and colleagues [45] studied expression patterns of the CRIP1a gene, *Cnrip1*, in *Xenopus laevis*, determining that mRNA is both temporally and spatially restricted during the early embryonic development. Quantitative RT-PCR using mRNA from different developmental stages indicated that *Cnrip1* expression follows a bimodal curve during development: *Cnrip1* was barely detectable until stage 10.5 at which time its expression increased to the first peak at stage 12 (late gastrula stage); after a decrease in stage 15, *Cnrip1* expression gradually increased to its second peak at stages 33/34 (late tailbud stage) [45]. These investigators used a loss-of-function approach with an antisense morpholino oligonucleotide designed to inhibit *Cnrip1* mRNA translation during development. *Cnrip1* morpholino oligonucleotides injected into embryos led to reduced head size, diminutive or absent eyes, and shortened or bent anterolateral axis, suggesting that the *Cnrip1* gene product is relevant for specific developmental stages [45]. *Cnrip1* mRNA knockdown resulted in reduced expression of four developmental transcription factors: PAX6 and RAX, which are early eye transcription factors; and SOX2 and OTX2, which are early neural transcription factors. Further studies with CRIP1a and CB_1_R homologs are necessary to attribute a role for the ECS in the *Cnrip1* mechanisms in *Xenopus* development.

Fin and colleagues [46] reported two *Danio rerio* (zebrafish) gene paralogs (*Cnrip1a* and *Cnrip1b*). These researchers used embryos at several different developmental stages to analyze the expression pattern in situ. *Cnrip1a* mRNA was noted throughout the embryo at 10 h post fertilization (hpf). At 24 and 48 hpf, *Cnrip1a* mRNA was most highly expressed in the brain and spinal cord [46], potential areas of developmental influence. Fin and colleagues used CRISPR/Cas9 genome editing to create *Cnrip1a* heterozygous knockouts in the zebrafish. The heterozygotes were crossed to produce *Cnrip1a* homozygotes. These *Cnrip1a*^−/−^ animals were produced at the expected frequency and exhibited no behavioral or morphological defects during 15 months of observation, suggesting that the *Cnrip1a* gene product is not necessary for proper morphological development, fertility, or viability [46]. Thus, the non-obligatory role of the *Cnrip1a* gene product on development in *Danio rerio* contrasts with the findings in *Xenopus laevis* [45]. This could be related to the differences in developmental regulation between fish and amphibians, or to differences in the structure/function of the *Cnrip1a* gene products between species.

Evidence suggests that CRIP1a may repress adult neurogenesis in mammalian species. Jung and colleagues [47], who were investigating the effects of pyridoxine (vitamin B6) on hippocampal function, found that CRIP1a protein was reduced 2-fold in hippocampi of pyridoxine-treated mice compared with control. To identify the relevance of reduced CRIP1a, these researchers silenced CRIP1a in male C57BL/6J mice at 8 weeks of age using bilateral injections of CRIP1a siRNA (versus scramble siRNA) into the dorsal hippocampi [47]. After 48 days, the CRIP1a knockdown in the hippocampi exhibited significantly more Ki67 positive nuclei and significantly greater density of immunodetectable doublecortin (DCX) [47]. Ki67 is a nuclear protein expressed throughout the cell cycle except during quiescence (G_0_) [48] and is a commonly used proliferation marker [49]. DCX is a neuronal migration protein associated with microtubules, which is expressed in neuroprogenitor cells during active cell division and remains in maturing neurons for two to three weeks [50]. The observed appearance of DCX and Ki67 upon silencing CRIP1a expression suggests that CRIP1a may suppress cellular proliferation and neurogenesis in the hippocampus. To assess whether the mechanism by which CRIP1a suppresses neurogenesis depends upon CB_1_R, Jung and colleagues [47] administered pyridoxine or saline vehicle (i.p., bid for 3 weeks), with or without a single dose of rimonabant (10 mg/kg, p.o.) at the beginning of the 3 weeks. Pyridoxine increased the number of Ki67 nuclei and immunodetectable DCX, but rimonabant treatment decreased both Ki67 and DCX regardless of pyridoxine treatment [47]. These data support a model whereby CRIP1a can suppress adult neurogenesis in a manner like CB_1_R antagonism.

The *Cnrip1* gene in mouse neuroprogenitor cells appears to undergo random monoallelic expression (also known as “allelic exclusion”) [51]. This means that *Cnrip1* may be expressed preferentially from a single chosen allele in a mosaic pattern whereby some cells express *Cnrip1* from the maternal chromosome, and others from the paternal chromosome. Monoallelic expression is based upon an epigenetic modification in autosomal genes analogous to gene inactivation in the X chromosome. Once selected, that epigenetic change remains stably propagated and is believed to contribute to cellular diversity and cell identity [51]. The mouse genome from neuroprogenitor clones differentiated from polymorphic F1 hybrid mouse embryonic stem cells was analyzed by RNA-seq to identify random monoallelic expression of autosomal genes. *Cnrip1* and *Plek* were found to be random monoallelic expression genes with overlapping 5′ ends. Using RNA FISH on cryostat sections of mouse embryos and adult brain, it was found that *Cnrip1* expression was monoallelic in the subventricular zone and the olfactory bulb [51]. In adult mouse neurons, CRIP1a expression is regulated post-transcriptionally by miR-128 [52]. The implications of these findings are not well understood, but perhaps there are epigenetic mechanisms of transcription and mRNA stability processes imparting CRIP1a cellular diversity across the central nervous system.

## 5. CRIP1a and Sensory Systems

A role of the ECS in the eye and the visual system has been identified (reviewed by [53]), and thus investigation of the potential for CRIP1a activity in the visual system and retina is merited. Using immunocytochemistry of ECS-related proteins, Hu and colleagues [54] found that CRIP1a in CD1 and C57BL/6 adult mice is expressed diffusely, stretching from the ganglion to the outer plexiform layer and terminating proximal to the outer nuclear layer containing photoreceptor cell bodies, as depicted in Figure 5. CRIP1a co-staining with CB_1_R appears near the outer plexiform layer [54]. However, these experiments may underestimate the cytochemical overlap of these two proteins because the CB_1_R antibody used in these experiments was directed against the rat CB_1_R C-terminus (amino acids 401–473) [54,55], a region predicted to be potentially occluded by CRIP1a. CRIP1a co-staining revealed proximal juxtaposition to diacylglycerol lipase-α (DAGLα), the postsynaptic 2-AG synthesizing enzyme [56,57,58], and postsynaptic density protein 95 (PSD95) [54], a membrane-associated guanylate kinase in retinal rod spherules and cone pedicles [59]. These experiments suggest that expression of retinal CRIP1a is primarily localized to the presynaptic termini (Figure 5). Outer plexiform retinal CRIP1a may be specific to cone, but not rod, photoreceptors, based upon evidence that DAGLα expression was limited to Type I OFF Cone Bipolar Cells, and proximity to PSD95 was observed only in cone terminals [54]. Alone, these experiments are sufficient to establish the presence of CRIP1a in glutamatergic cone cell termini, but insufficient to argue that CRIP1a expression in photoreceptor termini is limited to only those cells synapsing with Type I OFF Cone Bipolar cells. Furthermore, these synapses were not the only region of retinal CRIP1a expression. Within the inner nuclear layer and the plexiform layers surrounding it, CRIP1a co-staining exhibited both proximity and overlap with calbindin [54], a calcium binding protein expressed in the retina primarily within GABAergic horizontal and amacrine cells [60,61,62]. To summarize, retinal CRIP1a can be found presynaptically in both excitatory (glutamatergic cone cells) and inhibitory (GABAergic horizontal and amacrine cells) synapses and may (CB_1_R-dense outer plexiform layer) or may not (CB_1_R-deficient inner nuclear layer) be co-localized with CB_1_R (Figure 5). The CRIP1a expression across multiple retinal regions suggests a role in photoreceptor signal integration.

Regarding auditory sensory function, Lezirovitz and colleagues [65] found a genomic duplication in chromosome 2p14 that plays a role in autosomal dominant hearing loss. The DNA of interest includes two entire genes (*PLEK* and *CNRIP1*) as well as the first exon of PPP3R1 (protein coding in a regulatory subunit of Protein Phosphatase 3). These researchers performed a genetic family study and included a range of affected individuals from young adult to elderly to study the CRIP1a mRNA found in mononuclear cells from peripheral blood samples. CRIP1a mRNA levels were greater in those individuals with hearing loss than in those having normal auditory function. The anatomical distribution of CRIP1a was investigated in a mouse model by examining immunofluorescent CRIP1a in the cochleae and in situ hybridization of CRIP1a mRNA in spiral ganglion neurons [65]. CRIP1a expression was greater in the spiral ganglion neurons at the regions closer to the corti and those that were extending toward the brainstem. These data suggest that CRIP1a could contribute to hearing loss or developmental perturbation(s) that lead to hearing loss. Additional experiments are necessary to study involvement of both CB_1_R and CRIP1a in hearing loss.

## 6. CRIP1a in Neurophysiology and Seizures

Smith and colleagues [21] used whole-cell voltage-clamp recordings from autaptic neurons in culture that express ECS enzymes, CB_1_R and CRIP1a, as a model system to assess excitatory postsynaptic potentials (EPSP). Depolarization-induced suppression of excitation (DSE) and CB_1_R agonist (2-AG and methanandamide)-mediated reduction in EPSPs were both attenuated in cells expressing exogenous CRIP1a. The response to CRIP1a overexpression was believed to be CB_1_R-selective because EPSP changes mediated by an A1 adenosine receptor agonist were not affected by CRIP1a overexpression [21]. This contrasts with experiments reported by Guggenhuber and colleagues [30], in which CRIP1a was exogenously overexpressed in adult male C57BL/6N mice via bilateral injections of adeno-associated viral (AAV)-CRIP1a vector into the hippocampal CA1 regions. EPSPs in CA1 pyramidal neurons from hippocampal slices were detected using bipolar electrodes in the Schaffer collaterals, which synapse with CA1 pyramidal neurons. CRIP1a overexpression augmented HU210-mediated depression of EPSPs [30], but did not alter DSE. These experimental results argue for a physiological role for CRIP1a in modulating neuronal excitation, but much more data are needed to provide clarity regarding the influence of CRIP1a on neurotransmission.

The localization of CRIP1a in the forebrain and hippocampus [30] makes CRIP1a a potential candidate to modulate cannabinoid-mediated anti-convulsant activity. Focal temporal lobe epilepsy (TLE) and frontal lobe epilepsy are the first and second most common forms of epilepsy in humans, respectively [66]. The hippocampus is of interest due to its relative susceptibility to network excitation and convulsive agents [67], and the ease of modeling partial-type temporal lobe seizures through kainic acid (KA) induction [68,69]. KA is an excitatory amino acid that induces limbic seizures at nanomolar concentrations and neuronal loss at micromolar concentrations, similar to that observed in TLE patients [67,68,69]. The mechanism by which KA promotes these effects is by overstimulating activity in both excitatory and inhibitory neural pathways in the hippocampal formation [67]. Due to the abundance of excitatory collaterals synapsing at the CA3 region and the circuit loop design of the hippocampal formation, CA3 is often considered a ‘pacemaker’ for the limbic system. Therefore, the activation of CA3 pyramidal cells by mossy fiber kainate receptors, in addition to activation of CA3 glutamatergic collaterals, overwhelms the increased activation of inhibitory interneurons. The combined effect is excessive glutamatergic signaling despite enhanced inhibitory activity [67]. When KA-induced seizures progress to status epilepticus, the kainate receptor density increases on postsynaptic CA3 pyramidal cells and DG granule cells due to formation of novel aberrant mossy fiber synapses via collateral sprouting [70,71]. In this way, KA can be used to reduce the seizure threshold of the hippocampal formation, increasing its already high susceptibility to hypersynchronization of excitatory currents. Ben-Ari and Cossart [67] describe this phenomenon as ‘seizures beget seizures’.

In order to investigate the mechanism(s) for cannabinoid anticonvulsant activity in a KA model, Monory and colleagues [33] developed transgenic mice with selective CB_1_R deletions to assess seizure severity. After seizure induction by KA (30 mg/kg i.p.), mice were monitored for seizure activity every 15 min for 2 h, being scored for severity based upon tonic-clonic seizure characteristics [33,72,73]. Seizure severity increased in mice with site-defined loss of CB_1_R from the DG and CA1-3 regions and in mice with cell-specific loss of CB_1_R in glutamatergic, but not GABAergic, cells [33]. This suggests that CB_1_R expression in hippocampal glutamatergic neurons, specifically, provides moderate neuroprotection from KA-induced hypersynchronization activities. CB_1_R expression in GABAergic neurons may be necessary for short-term plasticity rather than seizure-related hyperactivity in pyramidal cells. Monory and colleagues [33] also conducted depolarization-induced suppression of inhibition (DSI) experiments in the same mouse lines using whole-cell voltage-clamp electrodes and glutamate receptor inhibitors to pharmacologically isolate GABAergic transmission. CB_1_R knockdown altered these currents, abolishing DSI [33]. This short-term plasticity is relevant in understanding general ECS mechanisms mediating neural activity but may be irrelevant to CB_1_R-related neuroprotection from seizure activity.

Guggenhuber and colleagues created adult male C57BL/6N mice that overexpressed CRIP1a by bilateral injections of AAV-CRIP1a vector into the dorsal hippocampal CA1 region, to compare with empty vector control mice expressing only endogenous CRIP1a [30]. These mice were administered KA (30 mg/kg, i.p.) and monitored for seizure activity every 15 min for 2 h using the protocol and scoring reported by Monory and colleagues [33]. Compared with controls, CRIP1a-overexpressing mice were rated a lower maximum severity score, and exhibited a non-statistically significant increase in seizure onset time [30]. Significantly fewer CRIP1a-overexpressing mice died from the KA injection compared with controls, suggesting that CRIP1a imparts neuroprotection against KA-induced excitotoxicity that may be sufficient to improve survival.

In order to better understand the cellular mechanism for the cannabinoid system effects on the KA model of seizures, Bojnik and colleagues [74] treated adult male Sprague-Dawley rats with KA (10 mg/kg, i.p.) and after 3 h, the animals were decapitated, the hippocampus dissected, and homogenates were analyzed for [^35^S]GTPγS binding. KA treatment produced statistically significant increases in the potency (EC_50_) and efficacy (E_max_) of ACEA-stimulated [^35^S]GTPγS binding to G proteins in the hippocampus, but not in the cortex [74]. In KA-treated mice, CRIP1a mRNA was significantly greater in both the hippocampus and cortex, whereas CB_1_R mRNA was moderately greater in the hippocampus but unchanged in the cortex [74].

Stauffer and colleagues [75] implicate CRIP1a as capable of selecting between at least two mechanisms involving the CB_1_R in neuroprotection against glutamate toxicity and cell death. These researchers administered a glutamate excitotoxic challenge (20 min, 300 μM glutamate) to primary neuronal cortical cultures from Long-Evans rats and determined that the neuroprotection imparted by WIN55212-2 was suppressed by CRIP1a overexpression from lentivirus transfer vector. However, CRIP1a overexpression was permissive to the ability of rimonabant to confer a statistically significant reduction in glutamate-induced cell death, a response not observed in empty vector-treated controls [75]. These experiments establish that CRIP1a selects for a CB_1_R inactivation neuroprotective mechanism against glutamate toxicity, over an existing CB_1_R activation mechanism. Further research may indicate that this is a promising area for designing pharmacotherapies for neuropathology.

To assess the significance of CRIP1a in human epilepsy, Ludanyi and colleagues [76] quantified and mapped ECS proteins in tissue samples of the hippocampal formation surgically removed via standard anterior temporal lobectomy from patients with therapy-resistant (intractable) TLE and control hippocampal samples dissected at autopsies of subjects who died from sudden, non-neurologic causes. qPCR of hippocampal mRNA revealed that TLE samples exhibited significantly lower expression levels of CRIP1a, CB_1_R, and DAGLα compared with controls; this was more pronounced in the sclerotic than non-sclerotic TLE samples. Immunohistochemistry and quantitative electron microscopy experiments further revealed significantly reduced CB_1_R density in the CA stratum radiatum, CA glutamatergic termini, and DG glutamatergic, but not GABAergic, termini of TLE patients compared to controls [76]. These data cannot be explained by differential loss of functional cells because the protein densities were normalized to housekeeping proteins β-actin and glyceraldehyde-3-phosphate dehydrogenase whose expression did not vary significantly between control, sclerotic, and non-sclerotic TLE samples. It is possible that repeated excitation seizure events, seizure-induced neuronal damage, or long-term protective measures against tissue damage associated with intractable epilepsy may have caused the downregulation of these proteins. Alternatively, an underlying low expression of these proteins could have reduced seizure threshold and might have been responsible for the development of epilepsy. Unfortunately, the confounding effects of medications could not be assessed as neither the previous nor current medication regimen for the patients were available. Regardless, CRIP1a is related to both pharmacologically-provoked seizures in animal model brains and recurrent seizure damage in the human epileptic brains. There is a need to include additional brain regions, seizure-induction model systems, and human epilepsy subtypes in future research. It would also be advantageous to incorporate seizure-susceptible animal models for analyzing cellular signaling in brain tissue.

## 7. CRIP1a and Neuropsychiatric Diseases

Schizophrenia is a neuropsychiatric disorder with heterogeneous genetic and neurobiological features, characterized by chronic or recurrent psychosis. The various behavioral manifestations of schizophrenic psychosis include positive symptoms (e.g., hallucination, delusion, disordered speech, paranoia and agitation), negative symptoms (e.g., anhedonia, speech poverty, social withdrawal and apathy), and cognitive impairments (e.g., memory and attention deficits). Schizophrenia has a poorly understood etiology and complex pathophysiology, but evidence implicates the ECS. For example, studies have found that schizophrenic patients are more frequently associated with cannabis use than patients with affective psychosis [77]. Using a convergent functional genomics approach, the *CNR1* gene encoding CB_1_R has been identified as a genetic factor involved in vulnerability to the development of schizophrenia [78]. The availability of CB_1_R is increased in the prefrontal cortex and anterior cingulate cortex in patients with schizophrenia [79,80,81,82]. A recent genome-wide DNA methylation analysis of human frontal cortex tissue from schizophrenic patients using the 450 Illumina array reports a significant decrease in DNA methylation at a CpG island in *CNRIP1* when compared to controls [83]. Despite the apparent role of CB_1_R, involvement of CRIP1a in the development of schizophrenia is largely unknown.

One neurodevelopmental model of schizophrenia involves gestational administration of an alkylating mitotoxin, methylazoxymethanol acetate (MAM), which disrupts mitosis and DNA methylation in specific regions of the developing brain [see reviews in [84,85]]. This method utilizes gestational MAM administration to specifically produce a development-disruption model of schizophrenia in rat offspring which exhibit behavioral, neurochemical and anatomical abnormalities comparable to those observed in individuals with schizophrenia [reviewed by [84]]. Perez and colleagues [86] found that the ventral hippocampi of F2 and F3 offspring from Sprague-Dawley rats treated with MAM on gestational day 17 exhibited increased numbers of spontaneously active dopaminergic neurons in the ventral tegmental area (VTA) and altered DNA methylation status of 181 different gene promotors, bi-directionally. Interestingly, one of the affected promoters was that of *Cnrip1* which exhibited decreased methylation in progeny of MAM-treated rats compared to saline controls [86]. Follow up experiments confirmed that this MAM-induced methylation decrease corresponded to increased CRIP1a protein expression. Notably, this did not alter CB_1_R expression levels and thus abolished the positive correlation between CB_1_R and CRIP1a expression found in control rats [87], consistent with previous findings [18] discussed above. Moreover, overexpression of CRIP1a by lentiviral-cDNA vector injection into the ventral hippocampus of male Sprague-Dawley rats elicited the neurological and behavioral changes consistent with the MAM model of schizophrenia: increased spontaneous activity of VTA dopaminergic neurons, impaired latent inhibition, and social avoidance [87]. These data implicate CRIP1a in the MAM-model and development of schizophrenia. It is not unreasonable, therefore, to propose that CRIP1a epigenetic control acts as a major player in the genetic predisposition to other complex neuropsychiatric pathologies which, like schizophrenia, possess both genetic and environmental factors. This opens the door for broad investigations into CRIP1a epigenetics. With regard to schizophrenia, future research should examine the mechanism(s) of CRIP1a and dopaminergic signaling in the VTA and overall mesolimbic pathway, areas strongly implicated in the pathophysiology of schizophrenia [see review in [88]].

In addition to schizophrenia, many psychiatric disorders have been implicated in the mesocorticolimbic pathway, such as Parkinson’s disease, substance use disorder, and attention deficit and hyperactivity disorder. For this reason, investigation into these regions is an important area of interest. The influence of CB_1_R and D_2_ dopamine receptor (D_2_R) expression on CRIP1a expression was investigated in an adult Sprague-Dawley rat model. Expression of the either of these receptors was reduced by AAV-delivery of receptor-specific siRNAs to the dorsal striatum [82]. Interestingly, striatal knockdown of either CB_1_R or D_2_R resulted in a sustained increase in CRIP1a mRNA and protein expression in the dorsal striatum. Increased CRIP1a was also found in striatal outflow nuclei (globus pallidus and entopeduncular nucleus), where the CB_1_R functions at the axonal terminals of GABAergic medium spiny neurons. In contrast, overexpression of CRIP1a in the dorsal striatum had no effect on levels of CB_1_R or D_2_R mRNA in the dorsal striatum or receptor protein levels in the striatum or outflow nuclei. Both CB_1_R knockdown and CRIP1a overexpression resulted in reduced ERK1/2 phosphorylation in the dorsal striatum and the outflow nuclei. Reduced striatal expression of mRNA for opioid peptides pro-enkephalin and pro-dynorphin produced by these medium spiny neurons was also observed [82]. This sequence of events preceded an increase in delta opioid receptor mRNA in the striatum, and opioid receptor protein in both striatum and the outflow nuclei [82]. These studies indicate that CRIP1a is a robust modulator of CB_1_R and D_2_R receptor-mediated functions as demonstrated in the basal ganglia. It is possible that CRIP1a might exert similar influences in other mesocorticolimbic regions of the brain.

CRIP1a was associated with protein changes that occur in the amygdala following chronic alcohol exposure [89]. Adult C57BL/6 mice were given access to ethanol in their home cage for 24 days using a two-bottle free choice paradigm in which water was in one bottle and ethanol (10% vol/vol) was in the other. The control animals had access to two water bottles. The ethanol-exposed animals displayed stable moderate alcohol intake (12.8 ± 1.3 g/kg/day from days 6 to 24), which did not alter the level of the stress hormone corticosterone and produced no dependence. A high-throughput screen of the amygdala (includes lateral, basolateral, central) proteome was performed using two-dimensional differential in-gel electrophoresis followed by matrix assisted laser desorption ionization tandem time-of-flight mass spectrometry. The results showed a 2-fold reduction in cytoplasmic CRIP1a protein in the amygdala in ethanol-drinking animals compared with controls [89]. However, there was no information provided on the CB_1_R for comparisons.

## 8. CRIP1a within Cancer Epigenetics

The evidence implicating CRIP1a epigenetics in the development of schizophrenia (described above) parallels observations of chromosomal modifications associated with genetic predisposition for a variety of cancers. The cannabinoid receptors have already joined the many possible targets for cancer treatment [90], making the recent report correlating *CNRIP1* chromosomal modifications on human chromosome 2p13 with cancer an unsurprising one. 

Sun and colleagues [91] reported that *CNRIP1* expression level inversely correlates with the expression of the gene for ATP binding cassette subfamily C member 3 (*ABCC3*) in glioma [91]. Overall survival and disease-free survival were greater in individuals with high *CNRIP1* and low ABCC3 expression levels. The rationale for this apparent linkage remains to be determined. Several studies have focused on the hypermethylation of the *CNRIP1* promotor and a small panel of other genes as a biomarker for colorectal cancer [92,93,94]. Lind and colleagues [92] analyzed a panel of 20 colon cancer cell lines and found that *CNRIP1* exhibited the highest level of methylation. Five other genes (*BEX1*, *C3orf14*, *FBN1*, *INA* and *SNCA*) were methylated in >80% of the cell lines. In contrast, methylation in normal cells was 0% for *CNRIP1*, *FBN1* and *INA* and <10% for *SNCA*. The methylation patterns in normal colon, adenoma (benign neoplasia) and colorectal cancer (malignant neoplasia) were then quantified for these genes and a few others. Importantly, in the adenoma and colorectal cells, hypermethylation correlated with their lower mRNA expression; normal cells had low levels of methylation and high gene expression. Treatment of the pre-cancerous and cancerous cells with 5-aza-2′deoxycytidine (5-azaC) and/or trichostatin A caused a reversal of methylation and increase in gene expression, lending additional support for the epigenetic regulation of *CNRIP1* and the use of this panel of genes as a screening tool. A study by Øster and colleagues [93] also found *CNRIP1* hypermethylation, but found correlations with a different set of genes containing highly frequent CpG islands. Hypermethylation of *CNRIP1* exons has also been demonstrated in colorectal cancer using an orthogonal LC-MS/MS-based technique [94]. It is important to note that hypermethylation of *CNRIP1* has also been observed in other cancer types: cholangiocarcinoma [95], lung adenocarcinoma [96], non-Hodgkin lymphoma [97], and gastric carcinoma [98,99].

Some of the downstream effects of *CNRIP1* promotor methylation in colorectal cancer have been determined. Zhang and colleagues [100] found that the methylation status of the CpG sequence at position 2245 correlated with invasion depth and TNM stage (tumor size, lymph node involvement and status of metastasis). Methylation of the promoter led to a decrease in mRNA and CRIP1a protein levels. Treatment of the cells with 5-azaC increased CRIP1a mRNA and protein levels but not to the levels of normal cells. SW620 colon cells, either treated with 5-azaC or transfected with a plasmid to overexpress CRIP1a, exhibited decreased proliferation. The same treatments also caused a decrease in invasiveness and migration ability, suggesting that CRIP1a is a tumor suppressor and that demethylation of the *CNRIP1* promoter could be a potential colorectal therapy. A more recent study by Sun and colleagues [101] tested the same hypotheses using siRNA knockdown or overexpression of CRIP1a. Knockdown of CRIP1a decreased proliferation but did not result in a significant difference in cellular migration and invasion in DLD1 and HCT8 cell lines. Overexpression of CRIP1a reduced proliferation as observed before [100]. There may also be a role for CRIP1a in pediatric acute myeloid leukemia (AML). Noort and colleagues present the RNA seq results from a collaborative study from an international cadre of pediatric oncology centers and their AML patients [102]. Many genes, including *CNRIP1*, exhibited a pattern of upregulation in patients with the *NUP98-KDM5A* background gene fusions (6.8-fold increase for *CNRIP1*) and downregulation in patients exhibiting the *NUP98-NSD1* background (2.4-fold decrease for *CNRIP1*). Although these observations help to stratify patients, the biological rationale or connection to AML for CRIP1a and the other genes identified is still unknown. Therefore, much work is still needed to understand the role of *CNRIP1*/CRIP1a in cancer.

A perusal of the available TCGA PanCancer Atlas within the cBIOPortal database (www.cbioportal.org) [103] indicates that the *CNRIP1* gene has 40 missense mutations in a number of tumor types: breast, colorectal, skin, esophageal, glioblastoma, kidney, liver, stomach, uterine, and ovary. Several truncating mutations and three gene fusions were also observed: brain, *CNRIP1*-*ZNF638*; ovary, *NEO1-**CNRIP1*; and lung, *PPP3R1*-*CNRIP1*. The latter was also reported by Galvan and colleagues [96], but the relationship between the Protein Phosphatase 3, regulatory subunit B,α (also known as calcineurin subunit B type 1) and CRIP1a remains to be determined [96]. Given the high degree of sequence identity across mammals (67–99%), it is certainly possible that the missense mutations may have compromised CRIP1a function, but this awaits further structural and biochemical study [104].

## 9. Summary and Prospectus

We are still in the early stages of uncovering the impact of the ECS which is crucial for maintenance of healthy physiology and is implicated in numerous pathologies, many seemingly disparate from each other. It is this complexity and apparent diversity that warrants careful interpretation of the nuances governing the ECS. Thus, untangling the relevant molecular mechanisms and designing therapeutic strategies necessarily requires comprehensive analysis of the cannabinoid receptor regulators. In this review, we have summarized available data on one CB_1_R regulator, CRIP1a, providing greater insight into the endocannabinoid influence on development as well as differentiated functions of cells.

Current data suggest that CRIP1a possesses broad suppressive activities, reducing CB_1_R available at the cell membrane and attenuating both constitutive and agonist-stimulated CB_1_R downstream effects. It furthermore appears capable of modulating the recruitment of other CB_1_R-associated proteins, such as Gi/o subtypes and β-arrestins. Curiously, the role of CRIP1a in the central nervous system may not be limited to CB_1_R modulation; its expression in CB_1_R-deficient cells and possible influence on other GPCRs suggests additional cannabinoid receptor-independent functions that have yet to be discovered. Putative physiological roles for CRIP1a include sensory systems regulation (vision and hearing) and neuronal excitability, particularly at the hippocampal formation. The physiological roles of CRIP1a at other brain regions (forebrain, cerebellum amygdala, and basal ganglia) have yet to be explained. Pathologies that have robust support for a CRIP1a mechanism include seizures, epilepsy, and schizophrenia. Additional pathologies or disorders (hearing loss, cancer, alcohol use disorder, and other psychiatric disorders), as well as developmental roles, have associations to CRIP1a that are less firm. All of these areas are excellent targets for further exploration, and both CB_1_R-dependent and -independent mechanisms should be considered in their analyses. Future explorations will require careful selection of model systems and a broader interpretation of the possible mechanisms available to CRIP1a. With enough prudence, we can expect to discover homeostatic processes regulated by CRIP1a that are crucial to human health.

## Figures and Tables

**Figure 1 biomolecules-10-01609-f001:**
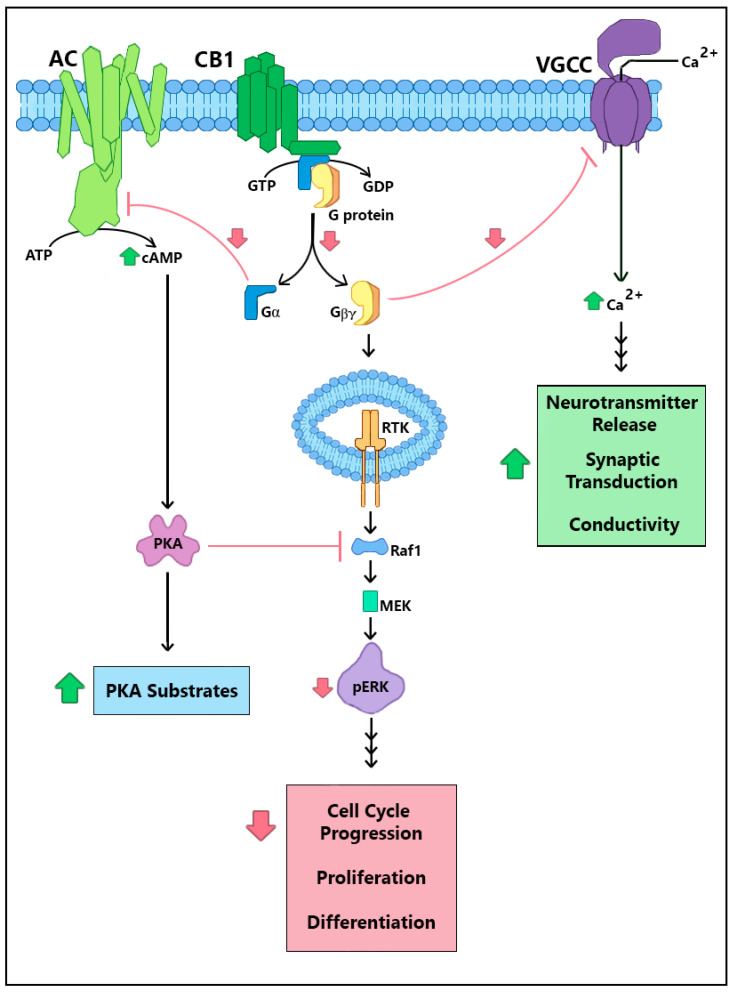
CB_1_R cellular signaling and points of CRIP1a-mediated suppression. Downstream effects of CB_1_R constitutive and agonist-mediated cascades include Gβγ-mediated inhibition of voltage-gated calcium channels (VGCC), Gαi-mediated inhibition of adenylyl cyclase (AC), and increased ERK1/2 phosphorylation through stimulation of receptor tyrosine kinase (RTK) and mitigation of PKA suppression. CRIP1a serves as a general negative-modulator, demonstrated to attenuate these basal and agonist-stimulated CB_1_R downstream actions: G protein activation, VGCC inhibition, cAMP diminution, and ERK1/2 phosphorylation. Green ‘up arrows’ reflect increased effects by CRIP1a expression; red ‘down arrows’ reflect decreased effects by CRIP1a expression.

**Figure 2 biomolecules-10-01609-f002:**
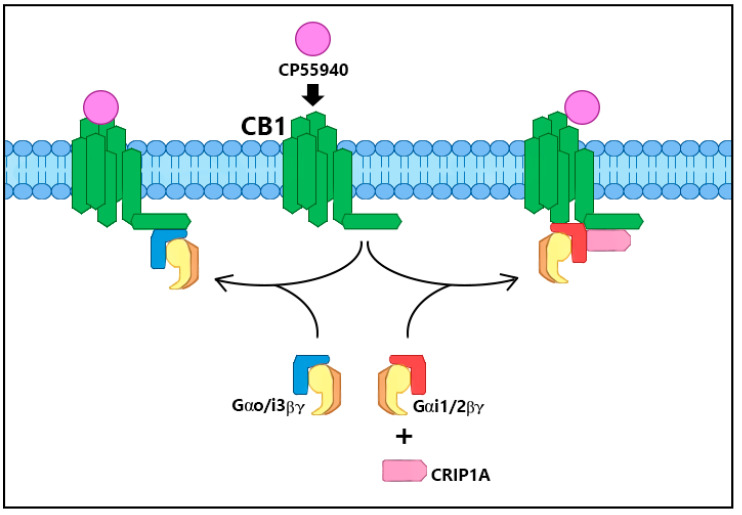
Selective G protein subtype switching. Under the influence of CRIP1a, CP55940-mediated Go/i3 activation by CB_1_R C is attenuated while Gi1/2 activation is promoted. The binding of Go/i3 to CB_1_R requires a region in the C-terminus, while Gi1/2 binding utilizes a region of the third intracellular loop. This G protein selection by CRIP1a can hypothetically alter the downstream cascades in response to CP55940 activation of CB_1_R depending upon the availability of these Gα subtypes.

**Figure 3 biomolecules-10-01609-f003:**
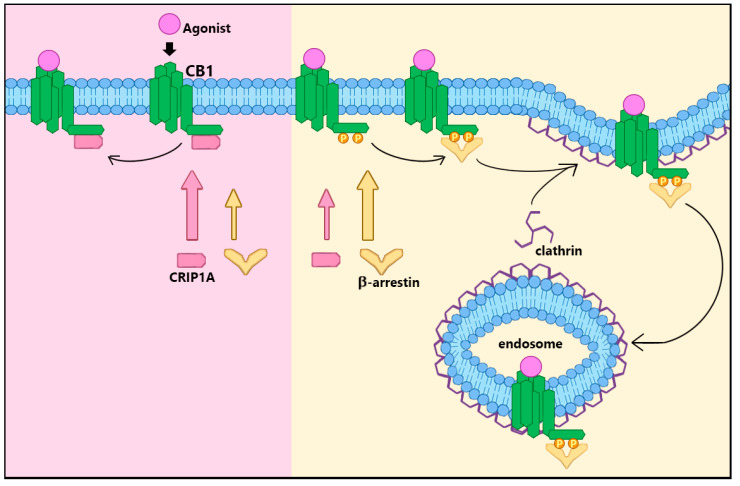
CRIP1a attenuation of β-arrestin/clathrin-dependent CB_1_R cell surface internalization. The mechanism of agonist-mediated, clathrin-dependent CB_1_R internalization from the cell membrane is dependent upon β-arrestin1/2 binding at the same C-terminal regions that bind CRIP1a. Preference for β-arrestin over CRIP1a is established by phosphorylation of specific Ser and Thr residues [20] within those binding domains. Preference for CRIP1a occurs when the same residues are not phosphorylated, attenuating agonist-stimulated CB_1_R internalization by precluding β-arrestin binding.

**Figure 4 biomolecules-10-01609-f004:**
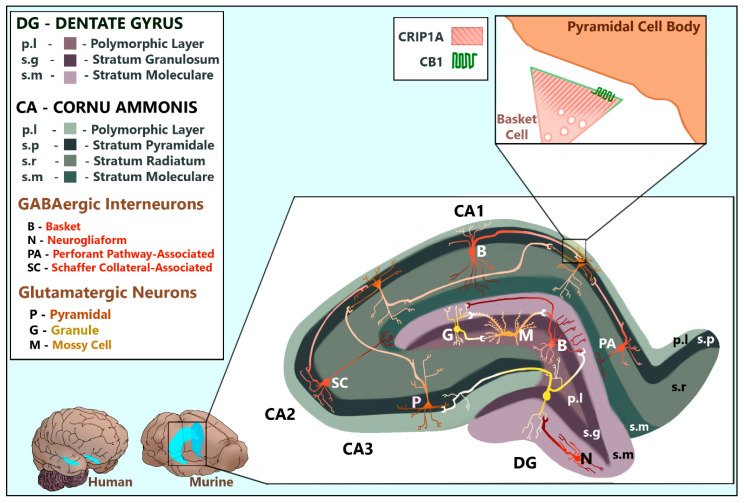
The mammalian hippocampal formation. CRIP1a is found dispersed in the presynaptic termini of GABAergic interneurons and glutamatergic neurons including DG granule cells, DG hilar mossy cells, and CA pyramidal cells. CB_1_R is localized to the presynaptic membranes of GABAergic interneurons and glutamatergic neurons including DG hilar mossy cells and CA pyramidal cells. This figure compares the relative size and location of the hippocampal formation in human and murine brains. The enlarged murine hippocampus depicts the shapes, positions, projections, and common synapses of relevant neurons within the layers of the dentate gyrus and hippocampus proper, viewed as a coronal slice. These cellular structures and positions were artistically rendered based upon data reviewed by [36,37,38,39,40]. The interneuron subtypes depicted were chosen based upon those with the strongest evidence of CB_1_R expression, as reviewed by Pelkey and colleagues [36]. The enlarged synapse depicts the subcellular localization of CB_1_R and CRIP1a in a CCK+ basket cell interneuron synapsing at a pyramidal soma, notably deficient of both proteins [36]. The synapse conformation was artistically rendered based upon data published by Dudok and colleagues [41].

**Figure 5 biomolecules-10-01609-f005:**
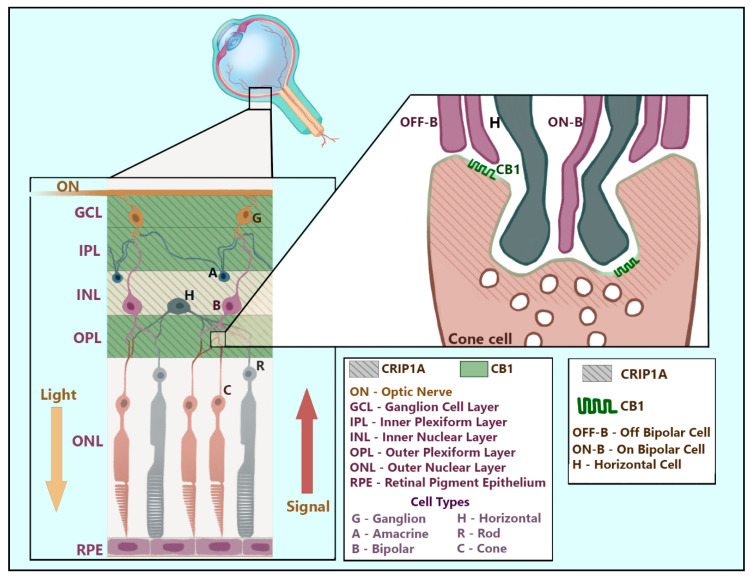
Distribution of CB_1_R and CRIP1a in the retina. CRIP1a (shown here as diagonal lines) is found dispersed throughout presynaptic regions of glutamatergic and GABAergic retinal neurons spanning the ganglion cell layer through the outer plexiform layer. CB_1_R is localized to the presynaptic membranes of retinal synapses in the ganglion cell layer and both inner and outer plexiform layers. This figure artistically renders the positions and common synapses of relevant neurons within the layers of the retina while superimposing identified regions of CRIP1a and CB_1_R expression. The enlarged synapse depicts the known region of CRIP1a and CB_1_R co-expression at the presynaptic zone of a cone cell in the outer plexiform layer. This artistic rendering was designed based on the described interaction between cone cells and bipolar cells by Chapot and colleagues [63] and on the shape and location of retinal neuronal cell types described in Neuroscience, 2nd Edition [64].

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
