# Peer review of "Cannabinoid Receptor Interacting Protein 1a (CRIP1a) in Health and Disease"

_biomolecules, 2020, doi:10.3390/biom10121609_

Round 1

Reviewer 1 Report

This is a very good and complete review, informative, detailed, synthetic, well written and with an essential content to know "cannabinoid receptor interacting protein 1a". Also, the article contains the appropriate references

I think that this article deserves publication.

However, I have some doubts that could be evaluated, without they are grounds for delaying acceptance of the paper.

  • I missed a brief comment on "cannabinoid receptor interacting protein 1b", a protein that the authors have also worked on.
  • The authors, I think that correctly, indicated that endocannabinoid agonists anandamide and 2-AG stimulate CB1R via orthosteric binding sites. After, they suggest a different behaviour between high and low efficacy antagonists. Do the authors know differences in binding sites (orthosteric and allosteric) of these ligands? If they know something about this aspect, some reference is known.
  • Could explain the different binding site, this different behaviour?

Author Response

Thank you very much for reviewing our manuscript! We have attached our revised version in response to all our reviewers’ comments. These edits to the written document are distinguished by red font color. Below we have listed the changes performed in response to your specific concerns and ideas:

(1) Unfortunately, there is very little literature on CRIP1b with the exception of 2 papers which have used molecular modelling to assess structure. Since our paper is centered on function and implications on health, and is not centered on structure, we thought it best to omit CRIP1b for this paper.

(2) We have removed the phrase “orthosteric binding sites” in order to prevent confusion. The information regarding various orthosteric and allosteric sites for small molecules on CB1R is a finely nuanced matter and we wanted to avoid confusion on the matter. These sites are distinct from larger protein binding sites. Two papers describing negative and positive small molecule allosteric modulators and their binding sites which are distinct from the CRIP1a binding site can be found below:

(1) N. Saleh, O. Hucke, G. Kramer, E. Schmidt, F. Montel, R. Lipinski, B. Ferger, T. Clark, P. W. Hildebrand, C. S. Tautermann, Angew. Chem. Int. Ed. 201857, 2580.

(2) Shao, Z., Yan, W., Chapman, K. et al. Structure of an allosteric modulator bound to the CB1 cannabinoid receptor. Nat Chem Biol 15, 1199–1205 (2019). https://doi.org/10.1038/s41589-019-0387-2

Reviewer 2 Report

The submitted manuscript is a comprehensive and detailed review article on Cannabinoid Receptor Interacting Protein 1 and describes its characterization, tissue distribution and functional roles in cannabinoid signlling in both health and disease. The manuscript is well organized and structured but sometimes the detailed description of single manuscripts distracts attention from the general context. However, the manuscript deserves attention and I only suggest minor improvements.

Specify in the text and not only in conclusion that the physiological roles of CRIP1a at brain regions other than hippocampus have yet to be deeply examined.

Figure 2: the representation of CRIP1a as diagonal lines is not very clear, especially in left panel in which diagonal lines are outside neuronal cells.

The paragraph on cancer seems to be quite out of the main focus of the manuscript which is very focused on the brain and related functions in health and disease. A better integration of this topic is suggested (e.g in introduction, inclusion of summary figure, etc).

In my opinion, the title of par 8 need to be changes: I suggest Epigenetic gene regulation of CNRIP1 implicated in cancer 

I suggest the inclusion of a figure in par 2 and an additional figure summarizing the main functional roles of the protein in health and disease.

Abbreviations need to be defined at the first appearance in the main text and used consistently all over the text. See for example “rimonabant” or endocannabinoid system.

Similarly avoid the repetition (e.g. CNRIP1 defined several time ad the encoding for.....; rimonabant defined several time as inverse agonist...); if necessary include a glossary

154 Explain the meaning of bold characters

Author Response

Thank you very much for reviewing our manuscript! We have attached our revised version in response to all our reviewers’ comments. These edits to the written document are distinguished by red font color. Below we have listed the changes performed in response to your specific concerns and ideas:

(1) We have included reference to the absence of data for roles of CRIP1a outside the hippocampus in section 3, lines 264-265.

(2) We have darkened the diagonal lines depicting CRIP1a localization in figure 5 (previously figure 2) which can be found on line 356.

(3) We have endeavored to improve the flow of the cancer section with the rest of the paper by relating the cancer epigenetic studies to similar chromosomal studies in the neuropsychiatric section. These edits can be found in section 7 lines 515-518 and section 8 lines 553-557.

(4) We have changed the title of paragraph 8 in a structure that is more consistent with the other titles in the review. It now reads “CRIP1a within Cancer Epigenetics”, line 552. Although this is not the precise change you suggested, we agree that it needed adjustment and we think this new change fits with the paper.

(5) We have included three new figures within section 2 in designed to depict important findings and conclusions from the information present in a manner that we believe best walks the reader through the narrative. The figures can be found on lines 128, 150, and 201 with their respective figure legends immediately below. We refer to those figures throughout the text, where appropriate, and made small structural or language changes in order to better explain those images (the largest of which can be found at lines 60-62, 45-46, and 188-195). Finally, we adjusted the figure numbers (4,5) when referencing the pre-existing figures, as appropriate.

(6) We considered the recommendation for a summary figure, but regretfully decided that we do not find it prudent to construct one at this time. Since there is still limited information regarding the full extent of possible human systems and pathologies CRIP1a is and is not involved in, creating a figure this early in the process may be more misleading than helpful. Hopefully our knowledge of CRIP1a will continue to grow so that, in the near future, we could create such a figure.

(7) We have combed through the paper and ensured that we have abbreviated and removed repetitive commentary where appropriate. Adjustments can be found at these lines: 79, 115, 170, 171, 557, 614. Deletions of repetitive/superfluous information were performed at these lines: 141, 290, 291, 294, 450, 493, 494, and 558.  

(8) We improved the explanation of the mimetic peptides and the bolded/underlined residues. Please see lines 175-182 and 188-195 for the full description, or lines 193-194 for the specific reference to bolded/underlined residues.

Reviewer 3 Report

This is a very good review about CRIP1a, a protein with multiple roles in the regulation of cannabinoid type 1 receptor (CB1R) function. The authors present a detailed description of the most important findings on CRIP1a and its physiological and putative pathological roles. Publication of this review will provide researchers with significant information. My only suggestion for the authors will be addition of a figure in the section 2, 'CRIP1a Cellular Mechanisms of Action' to visually represent how this molecule modulates CB1R expression and signaling.

Author Response

Thank you very much for reviewing our manuscript! We have attached our revised version in response to all our reviewers’ comments. These edits to the written document are distinguished by red font color. Below we have listed the changes performed in response to your specific concerns and ideas:

We have included three new figures within section 2 in designed to depict important findings and conclusions from the information present in a manner that we believe best walks the reader through the narrative. The figures can be found on lines 128, 150, and 201 with their respective figure legends immediately below. We refer to those figures throughout the text, where appropriate, and made small structural or language changes in order to better explain those images (the largest of which can be found at lines 60-62, 45-46, and 188-195). Finally, we adjusted the figure numbers (4,5) when referencing the pre-existing figures, as appropriate.
